# A Multigraph-based Model for Overlapping Entity Recognition

## Abstract

In this paper, we propose a new model that is capable of recognizing overlapping entities based on multigraphs, as opposed to simple graphs commonly used in graphical models for structured prediction. Through extensive experiments on standard datasets containing overlapping and non-overlapping entities, we demonstrate that our model outperforms previous models. We also present some analysis on the differences between our model and the previous models and discuss the possible implications of the differences. To the best of our knowledge, this is the first structured prediction model utilizing multigraphs to predict overlapping structures.

## 1 Introduction

Named entity recognition (NER), or in general the task of recognizing entities in a text, has been a research topic for many years (McCallum and Li, 2003; Nadeau and Sekine, 2007; Ratinov and Roth, 2009). However, as previously noted by Finkel and Manning (2009), many previous works ignored overlapping entities, although they are quite common. For example, the location entity *China* appears within the organization entity *Bank of China*. In practice, overlapping entities have been found in many existing datasets, including ACE (Doddington et al., 2004), GENIA (Kim et al., 2003), and in a clinical text dataset (Suominen et al., 2013). Figure 1 illustrates some examples of overlapping entities adapted from existing datasets.

Solving this task is non-trivial, as the typical way of modeling entity recognition as a sequence prediction problem (*e.g.*, using linear-chain CRF

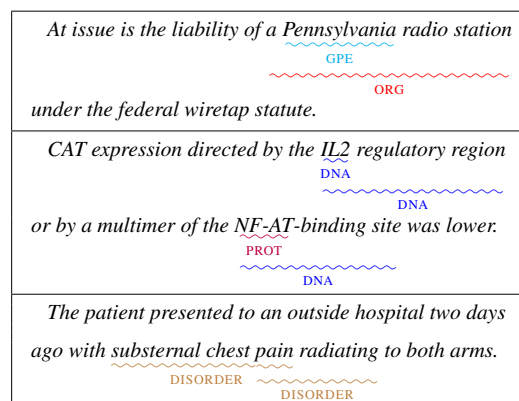

Figure 1: Examples of overlapping entities.

which utilizes simple graphs) has difficulties handling overlapping entities (Alex et al., 2007). Finkel and Manning (2009) proposed to use a tree-based constituency parsing model to handle nested entities with a time complexity that is *cubic* in $n$ for its inference procedure with $n$ being the number of words in the sentence. This complexity was later improved by Lu and Roth (2015) with a hypergraph-based model, which shows a time complexity that is *linear* in $n$.

In this work, as opposed to using simple graphs or hypergraphs, we introduce a novel *multigraph-based model* to tackle the problem of overlapping entity extraction. We show it is possible to assign explicit semantics to different edges connecting the same pair of nodes when representing structures, leading to the novel multigraph representations that can be used to represent overlapping structures. We present the training and inference procedures over such a novel representation. To the best of our knowledge, this is the first structured prediction model utilizing multigraphs to predict overlapping structures.

In this paper we make the following major contributions:

- We propose a novel multigraph-based model

for predicting overlapping entities.

- Empirically, we show that our model is able to achieve higher $F_1$-scores compared to previous models in multiple datasets.
- Theoretically, we show that unlike a previous state-of-the-art model that we compare against, our model does not present the *spurious structures* issue in its inference procedure and is therefore *nondeficient*. On the other hand, it still maintains the same inference time complexity as the previous model.

We also believe our proposed multigraph-based structured prediction framework can be used to solve other problems involving overlapping structures, and we hope this work can inspire further research along such a direction. We will make our system and code available for research purposes.

## 2 Related Work

Named entity recognition (NER) has been a research focus for quite some time. It is normally regarded as chunking task similar to base NP chunking (Kudo and Matsumoto, 2001; Shen and Sarkar, 2005), and hence the entities are usually represented in a similar way, using BILOU (Beginning of an entity, In the middle of an entity, Last token of an entity, Outside of any entity, Unit-length entity) or the simpler BIO annotation scheme (Ratinov and Roth, 2009). Some previous works have focused on getting the best representation of the chunks (Sang and Veenstra, 1999; Loper, 2007) and Ratinov and Roth (2009) showed that between BIO and BILOU scheme, the latter is the better approach for recognizing non-overlapping entities. As a chunking task, it is commonly modeled using sequence labeling models, such as the linear-chain CRF (Lafferty et al., 2001), which has time complexity $O(nT^2)$ with $n$ being the number of words in the sentence and $T$ the number of entity types.

On the task of recognizing entities that may overlap with one another, one of the earliest works that attempted to regard this task as a structured prediction task was McDonald et al. (2005). They represented entities as top-$k$ predictions with positive score from a structured multilabel classification model. This model resembles structured SVM (Tsochantaridis et al., 2005) in terms of the objective function used. Since they set $k = n$, their model has a time complexity of $O(n^3T)$.

Alex et al. (2007) proposed a cascading approach using multiple sequence labeling models, each handling a subset of all the possible entity types, where the models which come later in the pipeline have access to the predictions of the models earlier in the pipeline. For each entity type, they used a standard linear-chain CRF model to predict entities, resulting in the time complexity of roughly $O(nT)$ depending on how the pipeline is designed. However, they noted that "it involves extensive amounts of experimentation to determine the best model combination", due to the many possible ways to arrange the order of the pipeline. Their model only handles overlapping entities of different types.

Finkel and Manning (2009) later proposed a constituency parser to handle nested entities by converting each sentence as a tree, and each entity is represented as one of the subtrees. Their model has the standard time complexity for a binary parser: $O(n^3 |G|)$, where $|G|$ is the size of the grammar that is proportional to $T$ in the best case, and $T^3$ in the worst case. They showed that their model is able to outperform a semi-CRF baseline. By design, their model cannot handle crossing entities (overlapping entities which are not nested).

More recently, Lu and Roth (2015) proposed a hypergraph-based model called *mention hypergraphs* that is able to handle overlapping entities, yet it retains the linear time complexity $O(nT)$. The model was shown to achieve competitive results compared to previous models on standard datasets. As we will be making extensive comparisons against this pervious state-of-the-art model, we will briefly discuss this approach in the next section.

## 3 Mention Hypergraph

In the mention hypergraph model of Lu and Roth (2015), nodes and directed hyperedges[1] are used together to encode entities and their combinations. The following 5 types of states are used at the position $k$ of a sentence:

- $\mathbf{A}^k$ denotes all entities starting at $k$ or later
- $\mathbf{E}^k$ denotes all entities starting at $k$
- $\mathbf{T}_t^k$ denotes all entities of type $t$ starting at $k$
- $\mathbf{I}_t^k$ denotes all entities of type $t$ covering $k$
- $\mathbf{X}$ denotes the end of an entity (the leaf node)

Different hyperedges connecting these nodes are used to represent how the semantics of a node is composed from those of its child nodes.

---

[1] For brevity, in this paper we may also use *edge* to refer to *hyperedge* in some discussions.

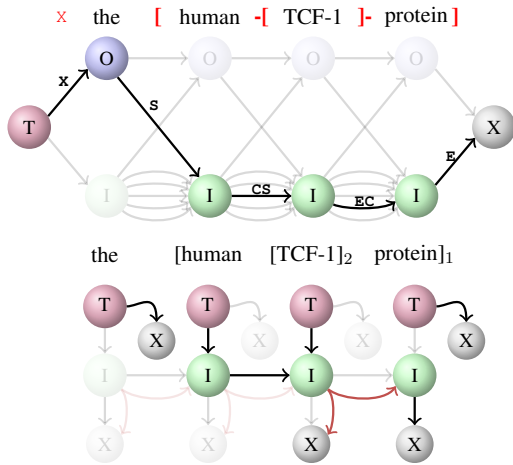

Figure 2: (Top) Our multigraph model representing overlapping entities, with the entity separators shown in red. (Bottom) The corresponding mention hypergraph ($\mathbf{A}$- and $\mathbf{E}$-nodes not shown).

Specifically, each $\mathbf{A}^k$ is connected to $\mathbf{A}^{k+1}$ and $\mathbf{E}^k$ through the hyperedge $\mathbf{A}^k \rightarrow (\mathbf{A}^{k+1}, \mathbf{E}^k)$, denoting the fact that the set of entities that start at $k$ or later is the union of the set of entities that start at $k+1$ or later and the set of entities that start at $k$. Each $\mathbf{E}^k$ is connected to $\mathbf{T}_1^k, \mathbf{T}_2^k, \ldots, \mathbf{T}_T^k$ through a hyperedge, denoting the fact that the entities that start at $k$ must be one of the $T$ types. Each $\mathbf{T}_t^k$ is connected to $\mathbf{I}_t^k$ through an edge (denoting there is an entity of type $t$ that starts at the $k$-th token) and to $\mathbf{X}$ through another edge (denoting there are no entities of type $t$ that start at the $k$-th token). Each $\mathbf{I}_t^k$ is connected to $\mathbf{I}_t^{k+1}$ (denoting there is an entity continuing to the next token), to $\mathbf{X}$ (denoting there is an entity ending here), or to both (with a single hyperedge, denoting the two cases above occur at the same time, a case of overlapping entities).

In this mention hypergraph, each possible entity is represented as a path from a $\mathbf{T}$-node to the $\mathbf{X}$-node through a sequence of $\mathbf{I}$-nodes (each denoting the words which are part of the entity), and the set of all entities present in a sentence forms a sub-hypergraph of the overall hypergraph.

The bottom part of Figure 2 shows how the mention hypergraph represents the two entities in the phrase "*the human TCF-1 protein*", which are "*TCF-1*" and "*human TCF-1 protein*" [2].

## 4 Multigraph-based Model

We now describe our proposed *multigraph-based model* for recognizing overlapping entities. In our

---

[2]Note that the word "*protein*" does not form an entity, due to the missing edge between the corresponding $\mathbf{T}$ node and $\mathbf{I}$ node.

| $w$   $w$ | $w$ [$w$ | $w$] $w$ | $w$] [$w$ |
|:---:|:---:|:---:|:---:|
| X | S | E | ES |
| $w$ - $w$ | $w$ -[$w$ | $w$]- $w$ | $w$]-[$w$ |
| C | CS | EC | ECS |

Figure 3: The 8 entity separators. The opening bracket ([), closing bracket (]), and dash (-) respectively refers to the three cases: next word starting an entity, current word ending an entity, and an entity continuation.

model, we assign for each token two states for each entity type, representing whether the token is part of an entity of a certain type. Formally, we define the following three types of nodes for each token:

- $\mathbf{T}_t$ denotes the set of entities of type $t$
- $\mathbf{I}_t^k$ denotes that the $k$-th token is part of an entity of type $t$
- $\mathbf{O}_t^k$ denotes that the $k$-th token is not part of any entities of type $t$

To represent the distinct roles a token can take, we consider the incoming edge and outgoing edge of that state. Unlike previous approaches, we assign explicit semantics to the edges in our multigraph-based model, which we call *entity separators*. We consider the 8 possible types of entity separators based on the combination of the following three cases:

1. An entity is starting at the next token (S)
2. An entity is ending at the current token (E)
3. An entity is continuing to the next token (C)

For each token, the possible combinations of cases are as follows: ECS, EC, CS, C, ES, E, S, and X, where X means none of the three cases applies. For example, the separator EC means there is one entity ending at the current token and another entity (overlapping) continuing to the next token. See Figure 3 for an illustration of these separators.

Next we define the edges between the states according to the 8 possible entity separators between adjacent tokens. Each entity separator is mapped to an edge connecting one state in the current position to another state in the next position depending on whether the separator defines current and next words as part of an entity, so in total we have 8 edges between two positions in the model. Some entity separators may connect the same two states, for example, the ES and C separator both connect $\mathbf{I}_t^k$ to $\mathbf{I}_t^{k+1}$ since in both cases the current token and the next token are part of an entity. In those cases, we simply define multiple edges between the pair of states, hence the *multigraph* aspect of

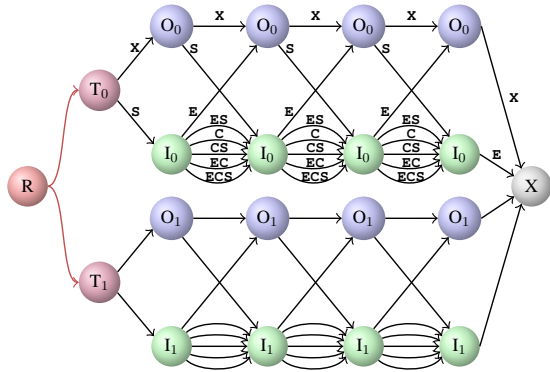

Figure 4: Example multigraph used in our model.

our model. The first **I**- and **O**-nodes in the sentence are connected to the **T**-node of the corresponding entity type, and the last **I**- and **O**-nodes are connected to the unique leaf node **X**. Finally, a designated root node **R** is added and is connected to all **T**-nodes by a single hyperedge to complete the multigraph structure. Figure 4 shows an example of our multigraph model considering 2 entity types and 4 input tokens. Note that the edges in our multigraph representations are directed, with nodes on the left serving as parents to the nodes on the right. Such directed edges will be helpful when performing inference, to be discussed in the next section.

The top part of Figure 2 shows how we can model the two entities "*human TCF-1 protein*" and "*TCF-1*" in the phrase "*the human TCF-1 protein*". Note how each edge maps to a distinct entity separator visualized in the text. Note how the roles of **T**-**I** edge and **I**-**I** edge in mention hypergraph are combined in our model using the single edge connecting the two **I** nodes for the token "*human*" and "*TCF-1*" to represent the entity separator CS.

### 4.1 Training, Inference and Decoding

We follow the log-linear approach to define our model, using the following regularized log-likelihood in training data $\mathcal{D}$ as our objective function:

$$\mathcal{L}_{\mathcal{D}}(\mathbf{w}) = \sum_{(\mathbf{x},\mathbf{y})\in\mathcal{D}} \left[ \sum_{\mathbf{e}\in\mathbf{y}} \mathbf{w}\cdot\mathbf{f}(\mathbf{e}) - \log Z_{\mathbf{w}}(\mathbf{x}) \right] - \lambda||\mathbf{w}||^2 \tag{1}$$

where $(\mathbf{x},\mathbf{y})$ is a training instance consisting of the sentence $\mathbf{x}$ and the correct output $\mathbf{y}$, $\mathbf{w}$ is the weight vector, $\mathbf{f}(\mathbf{e})$ is the feature vector defined over the edge $\mathbf{e}$, $Z_{\mathbf{w}}(\mathbf{x})$ is the normalization term, and $\lambda$ is the $l_2$-regularization parameter. The objective function is then optimized until convergence using L-BFGS (Liu and Nocedal, 1989).

We note the mention hypergraph model also defines the objective in a similar manner. For both models, the inference is done based on a generalized inside-outside algorithm. Both models involve directed structures, on top of which the inference algorithm first calculates the inside score for each node from the leaf node to root, and then the outside score from the root to the leaf node, in very much the same way as how inference is done in a classic graphical model. Specifically, for our multigraph-based model, the inside scores are calculated using a bottom-up (right-to-left) dynamic programming procedure, where we calculate the inside score at each node by summing up the scores associated with each path connecting the current node to one of its child nodes. Each such path score is in turn defined as the product of the inside score stored in that child node and the score defined over the edge connecting them. The computation of the outside scores can be done in an analogous manner from left to right. It can be verified that the time complexity of such an inference procedure for our model is $O(nT)$, which is the same as the mention hypergraph model.

During decoding, we perform MPE inference using a max-product procedure that is analogous to how the Viterbi decoding algorithm is used in conventional tree-structured graphical models to find out from the overall multigraph the highest-scoring subgraph, from which we extract entities[3].

## 5 Model Analysis

In this section we make two analytical comparisons between our model and the previous models.

### 5.1 State-based vs Edge-based Paradigm

Based on how previous works model the role of each token in defining the predicted entity spans, we can classify the previous works into two paradigms, namely state-based paradigm and edge-based paradigm, which we detail below.

**State-based Paradigm**

In conventional graph-based models for structured prediction such as linear-chain conditional random fields (Lafferty et al., 2001) and tree-based models (Finkel and Manning, 2009), typically nodes

---

[3]We note that both models involve interpretation of the output structures when recovering the entities. We implemented the same interpretation process as that was done in the mention hypergraph model, and we resolved ambiguous structures by considering them as nested entities instead of crossing entities.

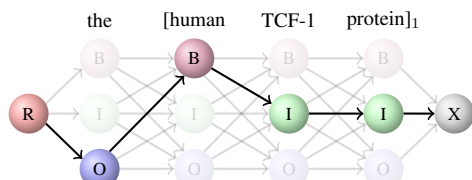

Figure 5: A linear-chain CRF model representing an entity in BIO scheme.

and edges are used together to encode structured output information. A linear-chain CRF model is shown in Figure 5 to model the protein entity "*human TCF-1 protein*" using the BIO scheme. In general, given an input sentence, a possible output can be represented using a linear-chain structure that involves states (nodes) and their connections (edges). One could observe that, in this case, given the state at each position in the structure, we already have enough information to interpret the output, without relying on the edge information. Similarly, this is also true for tree-based models. This leads to the following definition of what we call the ***state-based paradigm*** – the approach where the set of states alone can be used to interpret the predicted entities.

We observe that the majority of previous works fall into this paradigm (McDonald et al., 2005; Alex et al., 2007; Finkel and Manning, 2009; Tang et al., 2013).

**Edge-based Paradigm**

Modeling the roles each token can take as states is, however, not the only possible paradigm. In the *mention hypergraph* model by Lu and Roth (2015), they used only a small number of possible states for each token. Their approach does not model the roles that each token can take directly using the states only. In fact, in addition to states, they used edges (or hyperedges, and their combinations) to capture the complex combination of roles each token can take. This leads to an alternative paradigm called ***edge-based paradigm***: the approach where the set of states *together with* the edges are used to define the predicted entities.

To see the importance of edges in the edge-based paradigm, we show a simple illustrative example based on the hypergraph model of Lu and Roth (2015) with two words in Figure 6. The left graph and the right graph have exactly the same states. However, due to different edges (or hyperedges), these two graphs correspond to different entity combinations. Specifically, the left graph captures the set of non-overlapping entities $\{w_1,$

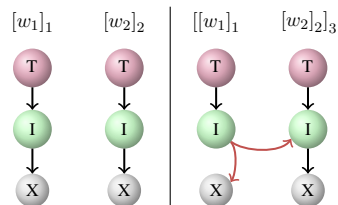

Figure 6: The importance of edges in the edge-based paradigm (in mention hypergraph model).

$w_2\}$, whereas the right graph captures the the following three entities: $\{w_1, w_2, w_1\ w_2\}$ with the third entity being overlapping with the first two.

It is easy to see that our multigraph-based model falls into this paradigm. Since there are multiple edges between the same pair of states, the combination of states alone do not define the roles of the tokens, and semantics are clearly assigned to the edges.

One might notice that there exists some equivalence between the two paradigms. For example, in our multigraph-based model, we can represent each edge (separator) as a separate state (such as a state named `ES` or `CS`) and assign such states to the gaps between two adjacent words using a linear-chain sequence labeling model, arriving at a "state-based" model. While this is feasible, we note that such an approach, however, will typically result in modeling unnecessary dependencies between the adjacent "states" (such as the dependencies between `ES` and `CS`, under the simple first-order assumption), leading to additional model complexity and computation overhead. Alternative approaches such as using nodes to replace the edges such as `ES` in our representation between adjacent **I** nodes are also possible. However, such approaches essentially exploit ad-hoc representations derived from the multigraph representation, making the model significantly less intuitive and less concise.

Since the edge-based paradigm is relatively unexplored compared to the state-based paradigm, we hope that this discussion can ignite further research in this direction.

## 5.2 Spurious Structures

In previous section we see that our model falls into the same edge-based paradigm as the mention hypergraph model. One might notice that our model is similar to the mention hypergraph model, in the sense that the edges in our model represent combinations of multiple edges in the mention hypergraph model. Since the objective function is de-

| | ACE2004 | | | ACE2005 | | | GENIA | | |
|---|---|---|---|---|---|---|---|---|---|
| | Train (%) | Dev (%) | Test (%) | Train (%) | Dev (%) | Test (%) | Train (%) | Dev (%) | Test (%) |
| # sentence | 6,799 | 829 | 879 | 7,336 | 958 | 1,047 | 14,836 | 1,855 | 1,855 |
| *w/ o.l.* | 2,685 (39) | 293 (35) | 373 (42) | 2,686 (37) | 341 (36) | 330 (32) | 3,199 (22) | 366 (20) | 448 (24) |
| # entities | 22,207 | 2,511 | 3,031 | 24,687 | 3,217 | 3,027 | 46,473 | 5,014 | 5,600 |
| *o.l.* | 10,170 (46) | 1,091 (43) | 1,418 (47) | 9,937 (40) | 1,192 (37) | 1,184 (39) | 8,337 (18) | 915 (18) | 1,217 (22) |

Table 1: Statistics of the datasets used in this experiments. *w/ o.l.*: sentences containing overlapping entities; *o.l.*: overlapping/nested entities.

fined using features over the edges, this raises the question: are the two models actually optimizing the same objective function?

To answer this question, let us look into the normalization term $Z_{\mathbf{w}}(\mathbf{x})$ calculated by each model using the inside-outside algorithm. Let $\beta(m)$ denote the inside score of the node $m$. Recall that in the inside-outside algorithm, the inside score of a node $m$ represents the sum of the scores of all paths from $m$ to the leaf node. Since in mention hypergraph a path from a $\mathbf{T}$-node to the leaf node represents an entity, $\beta(\mathbf{T}_t^k)$ represents the scores of all entities starting at position $k$ of type $t$. Similarly, $\beta(\mathbf{E}^k)$ represents the scores of all entities starting at position $k$ of any type. Now, since $\beta(\mathbf{A}^k)$ is calculated as the product between $\beta(\mathbf{A}^{k+1})$ and $\beta(\mathbf{E}^k)$, it represents the score of all combinations of entities starting at $k$ or later. Therefore, the normalization term $Z_{\mathbf{w}}(\mathbf{x}) = \beta(\mathbf{A}^0)$ represents the score of *all combinations of entities* in the sentence.

Consider a sentence with three words $w_0$, $w_1$, $w_2$. The normalization term includes the scores of both these distinct entity combinations:

1. Entities $w_0 w_1$ and $w_1 w_2$ (crossing entities)
2. Entities $w_0 w_1 w_2$ and $w_1$ (nested entities)

However, as also noted by Lu and Roth (2015)[4], these two combinations share the same subgraph in the model.[5] On the other hand, in the mention hypergraph model, only one of the many combinations would be considered in the first term of the objective of equation $1 - \sum_{\mathbf{e} \in \mathbf{y}} \mathbf{w} \cdot \mathbf{f}(\mathbf{e})$, whose score can be calculated efficiently using the inside algorithm. The other structures that share the same subgraph will become what we call *spurious structures* – the structures that would never be predicted by the model. Also note that the score of each of these distinct interpretation of the same sub-hypergraph might differ from each other. Refer to the supplementary material for more details. We call a model with normalization term $Z_{\mathbf{w}}(\mathbf{x})$

that contains spurious structures a *deficient* model.

In contrast, we can observe that, unlike the mention hypergraph model, our model is *nondeficient* – the normalization term $Z_{\mathbf{w}}(\mathbf{x})$ calculated by our multigraph-based model does not contain spurious structures. This can be verified by the fact that the inside score of each node in our multigraph is calculated as the sum of the scores associated with each path following that node.

## 6 Experiments

### 6.1 Datasets

To assess our model's capability in recognizing overlapping entities and make comparisons with previous models, we looked at datasets where overlapping entities are explicitly annotated. Specifically, we looked at the following standard datasets which were used in several previous works (Finkel and Manning, 2009; Lu and Roth, 2015): ACE2004, ACE2005, and GENIA. For ACE datasets, we used the same splits as used by Lu and Roth (2015) published in their website[6]. For GENIA, we used GENIAcorpus3.02p[7] that comes with POS tags for each word (Tateisi and Tsujii, 2004). Following previous works (Finkel and Manning, 2009; Lu and Roth, 2015), we first split the last 10% of the data as the test set. Next we used the first 80% and the subsequent 10% for training and development, respectively. We made the same modifications as described by Finkel and Manning (2009) by collapsing all *DNA*, *RNA*, and *protein* subtypes into *DNA*, *RNA*, and *protein*, keeping *cell line* and *cell type*, and removing other entity types, resulting in 5 entity types. The statistics of each dataset is shown in Table 1. We can see overlapping entities are common in such datasets.

### 6.2 Features

For both models (mention hypergraph and our model) that fall under the edge-based paradigm, we define features over the edges in the models.

---

[4]See Figure 3 in Section 3.2 in their paper.

[5]In fact, there are 7 distinct entity combinations which have the same subgraph.

[6]http://statnlp.org/research/ie#mention-hypergraph

[7]http://geniaproject.org/genia-corpus/pos-annotation

|  | ACE2004 | | | ACE2005 | | |
|---|---|---|---|---|---|---|
|  | $P$ | $R$ | $F_1$ | $P$ | $R$ | $F_1$ |
| Lin-CRF | 70.3 | 42.0 | 52.6 | 67.0 | 45.6 | 54.3 |
| MH | **80.7** | 46.1 | 58.7 | **77.6** | 47.3 | 58.8 |
| MH (*) | 71.0 | 55.4 | 62.2 | 67.0 | 57.3 | 61.7 |
| Ours | 76.3 | 52.0 | 61.9 | 73.5 | 51.6 | 60.6 |
| Ours (*) | 71.6 | **58.3** | **64.3** | 67.5 | **59.1** | **63.0** |

Table 2: Results on ACE2004 and ACE2005.

|  | $P$ | $R$ | $F_1$ |
|---|---|---|---|
| Semi-CRF | 76.2 | 61.7 | 68.2 |
| F&M ('09) | 75.4 | 65.9 | 70.3 |
| MH | **80.6** | 58.6 | 67.8 |
| MH (*) | 75.0 | 66.1 | 70.3 |
| Ours | 80.1 | 61.6 | 69.7 |
| Ours (*) | 74.6 | **67.2** | **70.7** |

Table 3: Results on GENIA dataset.

Features are defined as string concatenations of input features – information defined over the inputs (such as current word and POS tags of surrounding words) and output features – structured information defined over the output structure. In our model, when an edge represents a combination of roles, a feature is defined for each output feature. This allows us to use to make use of the identical set of features for both our multigraph model and the baseline mention hypergraph model, in order to make a proper comparison. We also followed Lu and Roth (2015) to add the additional *mention penalty* feature defined over the edges that represents the start of an entity to tune $F_1$-scores on the development set.

When defining the input features for both our model and the mention hypergraph model, we tried to follow as close as possible to the features used by previous works in each dataset: we followed Lu and Roth (2015) for the features used in ACE datasets, and Finkel and Manning (2009) for features used in GENIA dataset. In general, they include surrounding words, surrounding POS tags, bag-of-words, Brown clusters (for GENIA only), and orthographic features. See the supplementary material for more details on the features.

### 6.3 Experimental Setup

We trained each model in the training set, then tuned the $l_2$-regularization parameter based on the development set from the values in {0.0, 0.001, 0.01, 0.1, 1.0}. For GENIA experiments, we also tuned the number of Brown clusters from the values in {100, 1000}. Following (Lu and Roth, 2015), we also used each development set to tune the mention penalty to optimize the $F_1$-score and report the scores on the corresponding test sets separately. Similar to Finkel and Manning (2009), as another baseline model we also trained a standard linear-chain CRF using BILOU scheme. Although this model does not support overlapping entities, it gives us a baseline to see the extent to which our model's ability to recognize overlapping entities can help the overall performance.

## 7 Results and Discussion

Table 2 and 3 show the results on the ACE and GENIA datasets, respectively. Following previous works (Finkel and Manning, 2009; Lu and Roth, 2015), we report standard precision ($P$), recall ($R$) and $F_1$ percentage scores. For the ACE datasets, we make comparisons with the linear-chain CRF baseline (Lin-CRF), which does not support overlapping entities, as well as our implementation of the mention hypergraph baseline (MH). For the GENIA experiments, besides our implementation of the mention hypergraph baseline, we also make comparisons with the results of the two systems reported in (Finkel and Manning, 2009) – a model based on semi-Markov CRF (Semi-CRF) that cannot handle overlapping entities and their proposed constituency tree based model (F&M ('09)) that can support overlapping entities. The (*) marks the results after optimizing the $F_1$-score in development set. Best results are highlighted in bold.

From such empirical results we can see that our proposed multigraph-based model yields significantly better results than the mention hypergraph model, with the highest improvement in $F_1$ of 2.1 points observed in ACE2004. In GENIA dataset, our implementation of mention hypergraph matches the performance of the constituency parser-based model of Finkel and Manning (2009), while our multigraph-based model outperforms both by about 0.4 point. As expected, the linear-chain CRF baseline yields relatively lower results compared to the other models, since it cannot predict overlapping entities. However, such results give us some idea on how much performance increase we can gain by properly recognizing overlapping entities.

As these datasets consist of both overlapping and non-overlapping entities, to further understand the effectiveness of each model in recognizing overlapping entities (and non-overlapping entities), we performed some additional experiments. Specifically, we split the test data into two portions, one that consists of only sentences that

| | % | MH (*) | | | Ours (*) | | |
|---|---|---|---|---|---|---|---|
| | | P | R | $F_1$ | P | R | $F_1$ |
| ACE2004 w/ o.l. | 42 | 72.8 | 51.0 | 60.0 | 71.3 | 56.2 | 62.8 |
| ACE2004 w/o o.l. | 58 | 73.1 | 62.8 | 67.5 | 72.5 | 64.2 | 68.1 |
| ACE2005 w/ o.l. | 32 | 68.6 | 52.8 | 59.7 | 68.5 | 55.7 | 61.4 |
| ACE2005 w/o o.l. | 68 | 64.8 | 64.8 | 64.8 | 66.0 | 64.7 | 65.4 |
| GENIA w/ o.l. | 24 | 77.1 | 60.3 | 67.7 | 75.6 | 60.7 | 67.3 |
| GENIA w/o o.l. | 76 | 73.9 | 70.1 | 71.9 | 74.1 | 71.6 | 72.9 |

Table 4: Scores on different types of sentences.

| | Dev | | | Test | | |
|---|---|---|---|---|---|---|
| | P | R | $F_1$ | P | R | $F_1$ |
| Lin-CRF | 90.0 | **88.9** | 89.4 | 84.2 | **83.6** | 83.9 |
| Illinois NER | - | - | 89.3 | - | - | 83.7 |
| MH | **94.7** | 83.1 | 88.5 | 91.1 | 76.8 | 83.3 |
| MH (*) | 91.8 | 87.1 | 89.4 | 87.2 | 80.9 | 83.9 |
| Ours | 94.6 | 84.3 | 89.1 | 91.0 | 78.0 | 84.0 |
| Ours (*) | 92.3 | 87.1 | **89.6** | 87.8 | 81.0 | **84.3** |

Table 5: Results on CoNLL 2003.

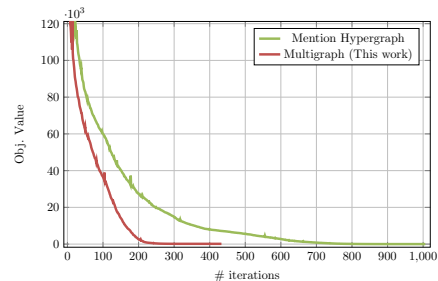

Figure 7: Objective vs. training iterations.

contain at least one entity that overlaps with another (w/ *o.l.*), and the other consisting of sentences that do not contain any overlapping entity (w/o *o.l.*). The results are shown in Table 4.

We can see that in ACE datasets, our model achieves higher $F_1$-score compared to the mention hypergraph for both portions, but it achieves slightly lower results in GENIA dataset for the portion that contains overlapping entities. Noticing that GENIA dataset has relatively lower percentage of overlapping entities compared to ACE, we believe that this low recall is due to insufficient amount of training data on overlapping entities for this dataset, which affected the models' confidence in predicting overlapping entities. When given sufficient training data on overlapping entities, which is the case for ACE, our model is able to better recognize overlapping entities.

Such results also lead to the interesting empirical finding that our model appears to be able to do well on the task of recognizing non-overlapping entities. As a further investigation, we performed yet one additional set of experiments, on the standard CONLL2003 dataset, which consists of only entities that do not overlap with one another.

The results are shown in Table 5. Again we see that our multigraph model outperforms all baseline models, including the Illinois NER system where external resources are not used (Ratinov and Roth, 2009), and a linear-chain CRF model, although the linear-chain CRF baseline models some interactions between distinct entity types and our model does not. Such results also suggest that modeling the interactions between distinct entity types may not be crucial for achieving a good performance in entity recognition. By examining the outputs, we found that although the training set is not annotated with overlapping entities, our model (and mention hypergraph model) will be able to predict overlapping entities when it is confident, leading to improved performance.

When comparing the basic version of our model (without optimizing $F_1$) against that of the men-

tion hypergraph model, we note that our model consistently yields a higher recall. We speculate this is due to the fact that as a nondeficient model that resolves the issue of spurious structures we discussed in Section 5.2, our model is more confident in making its predictions.

We also empirically analyzed the training speed and convergence properties of the two models. Empirically, as illustrated in Figure 7 which shows how the objective improves when the training progresses on ACE2004, we found our multigraph-based model requires significantly less iterations to converge than mention hypergraph, though each training iteration requires 12% more time.

# 8 Conclusion and Future Work

We presented a novel *multigraph-based model* for entity recognition from text where entities may overlap with one another. We showed that empirically our model is able to yield better recognition results compared to previous models. We also performed theoretical analysis on the model and showed that our model resolves the *spurious structures* issue associated with a previous state-of-the-art model, while still maintaining the same inference time complexity.

Future work includes further investigations on how to apply the proposed multigraph-based formalism to other structured prediction problems involving complex structures, as well as finding applications of the proposed model in other related NLP tasks that involve the prediction of overlapping structures, such as equation parsing (Roy et al., 2016).

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

## A Details on Spurious Structures

About mention hypergraph, we remarked that the normalization term includes spurious structures. This section shows in more details how this is the case using some examples.

Consider the simplified mention hypergraph as shown in Figure 8 (left) consisting of three words and where the possible edges have been restricted to what are shown in the figure. Namely, there are only three possible subgraphs here, one for each of the three (hyper-)edges coming out from the node $\mathbf{I}_1$ associated with the word "Apache". Let A, B, C, D, E, F respectively denote the edges $\mathbf{T}_1 \rightarrow (\mathbf{I}_1)$, $\mathbf{I}_0 \rightarrow (\mathbf{I}_1)$, $\mathbf{I}_1 \rightarrow (\mathbf{I}_2)$, $\mathbf{I}_1 \rightarrow (\mathbf{X})$, $\mathbf{I}_1 \rightarrow (\mathbf{I}_2, \mathbf{X})$, and $\mathbf{I}_2 \rightarrow (\mathbf{X})$ as shown in Figure 8 (left). Further assume that features are only defined on these labeled edges.

Now we consider the normalization term $Z_{\mathbf{w}}(\mathbf{x}) = \beta(\mathbf{A}_0)$ calculated by inside-outside algorithm on this graph. The inside score $\beta(\mathbf{I}_1)$ is the sum of the scores of the three possible entity combinations that include the word "Apache" as the first token in the entities, namely: {*Apache*} (using the edge $\mathbf{I}_1 \rightarrow (\mathbf{X})$), {*Apache helicopter*} (using the edge $\mathbf{I}_1 \rightarrow (\mathbf{I}_2)$), and {*Apache helicopter*, *Apache*} (using the hyperedge $\mathbf{I}_1 \rightarrow (\mathbf{I}_2, \mathbf{X})$). Let us call this set of three entities as $S_1$. The inside scores of $\mathbf{I}_0$, $\mathbf{A}_1$, the $\mathbf{T}$-nodes, the $\mathbf{E}$-nodes will be the same as this as there are no features along the unique edges between these

nodes and the node $\mathbf{I}_1$. Note that the inside score $I(\mathbf{T}_0)$ represents the set of entities $S_2$ which is the same as $S_1$ except that each entity also includes the word "an" as the first word.

Now, the normalization term $\mathbf{A}_0$ includes the 9 possible entity combinations which are the results of taking all possible combinations between $S_1$ and $S_2$. The list of paths used to represent each of these combinations is as follows (only part of the paths is shown, the rest is implied as there is only one way to connect to the root node $\mathbf{A}_0$):

1. A-C-F and B-C-F (*Apache helicopter*, *an Apache helicopter*)
2. A-C-F and B-E-F (*Apache helicopter*, *an Apache helicopter*, *an Apache*)
3. A-C-F and B-D (*Apache helicopter*, *an Apache*)
4. A-E-F and B-C-F (*Apache helicopter*, *Apache*, *an Apache helicopter*)
5. A-E-F and B-E-F (*Apache helicopter*, *Apache*, *an Apache helicopter*, *an Apache*)
6. A-E-F and B-D (*Apache helicopter*, *Apache*, *an Apache*)
7. A-D and B-C-F (*Apache*, *an Apache helicopter*)
8. A-D and B-E-F (*Apache*, *an Apache helicopter*, *an Apache*)
9. A-D and B-D (*Apache*, *an Apache*)

Note that except the first one (the $\mathbf{X}$-node associated with $\mathbf{I}_1$ is missing) and the last one ($\mathbf{I}_2$ and the last $\mathbf{X}$-node are missing), the other 7 entity combinations are actually represented using the same subgraph consisting of the edges {A, B, E, F}, which is what is calculated in the fifth item (A-E-F and B-E-F). The other combinations represent entity combinations which, while valid, are not represented in the model as such, and so never appear as numerator in the likelihood. This is because whenever that entity combination appears, the model will represent it using the subgraph shown in Figure 8 (right).

## B Features

For ACE datasets we used these features:

1. Words and POS tags (with window of 3 words to the left and right of current word)
2. Words and POS tags n-gram (up to length 4 containing current word)
3. Bag-of-word (with window of 5 words to the left and to the right of current word)
4. Orthographic (following Lu and Roth (2015))

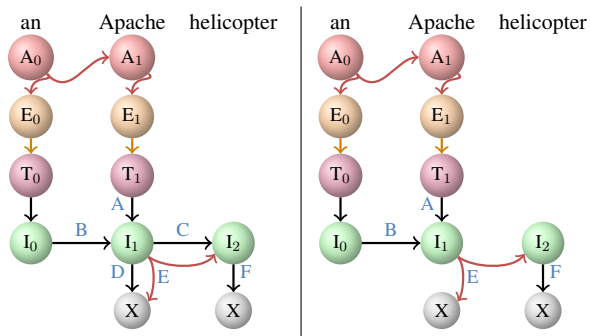

Figure 8: (left) A simplified example of a mention hypergraph with restricted edges. (right) The subgraph associated with 7 out of 9 possible entity combinations.

5. Parent node type

For GENIA we used these features:

1. Words and POS tags (with window of 2 words to the left and right of current word)
2. Words and POS tags n-gram (up to length 4 containing current word)
3. Bag-of-word (with window of 5 words to the left and right of current word)
4. Brown clusters with window of 1 word to the left and right of current word (using 100 or 1000 clusters built from training data only)
5. Word shape (Finkel and Manning (2009))
6. Prefixes and suffixes of current word (up to length 6)
7. Edge type

For CoNLL 2003 we used these features:

1. Words and POS tags (with window of 2 words to the left and right of current word)
2. Words and POS tags n-gram (up to length 4 containing current word)
3. Bag-of-word (with window of 5 words to the left and right of current word)
4. Word shape (with window of 2 words to the left and right of current word)
5. Prefixes and suffixes of current word (up to length 5)
6. Orthographic (following Lu and Roth (2015))
7. Edge type

## C  Hyperparameter

Table 6 lists the best $l_2$-regularization coefficient $\lambda$ for each dataset and model. For GENIA, the

|  | ACE2004 | ACE2005 | GENIA | CoNLL |
|---|---|---|---|---|
| MH | 0.001 | 0.0 | 1.0 | 0.01 |
| MH (*) | 0.001 | 0.1 | 1.0 | 0.001 |
| Ours | 0.001 | 0.001 | 1.0 | 0.001 |
| Ours (*) | 0.1 | 0.1 | 1.0 | 0.001 |

Table 6: The value of $l_2$ regularization parameter that gives the best result in development set.

optimal Brown cluster size was found to be 1000, except for 'Ours (*)', where the best cluster size is 100.

## D  GENIA Preprocessing

For GENIA, we used GENIAcorpus3.02p that comes with POS tags for each word (Tateisi and Tsujii, 2004). Similar to the problem faced by Finkel and Manning (2009) on JNLPBA dataset, we also find tokenization issues in this corpus. As described by Tateisi and Tsujii (2004), when a hyphenated word such as *IL-2-induced* is partially annotated as an entity (in this case *IL-2*), the POS annotation corpus splits it into two tokens, which when done in test set will leak some information about the presence of entity. Unlike Finkel and Manning (2009) which tried to match the tokenization during testing, we simply further split all tokens at some punctuations (those matching the regular expression [-/,.+]), while keeping the information that they all originally come from the same word. This has the advantage of simplifying the tokenization procedure, although it makes the task slightly more difficult due to the higher number of tokens.

Also, to handle the discontiguous entities present in GENIA dataset (mainly due to coordinated entities involving ellipsis), following the approach used by the JNLPBA shared task organizer (Kim et al., 2004), we consider a group of coordinated entities as one structure. For example, in "... *the [T- and B-lymphocytes] count in* ...", the entities "*T-lymphocytes*" and "*B-lymphocytes*" are annotated as one structure "*T- and B-lymphocytes*".

