# Peer review of "A Multigraph-based Model for Overlapping Entity Recognition"

_ACL 2017 — decision unknown_

[Official Review · Reviewer 1 · rating 3 · confidence 3]
soundness 4 · originality 3 · clarity 3 · impact 3 · substance 3 · appropriateness 5 · meaningful comparison 2 · presentation format Poster

- Strengths: the paper is well-written, except for a few places as described
below. The problem the paper tackles is useful. The proposed approach,
multigraph-based model, is a variant of MH. The empirical result is solid.

- Weaknesses: Clarification is needed in several places.

1. In section 3, in addition to the description of the previous model, MH, you
need point out the issues of MH which motivate you to propose a new model.

2. In section 4, I don't see the reason why separators are introduced. what
additional info they convene beyond T/I/O?

3. section 5.1 does not seem to provide useful info regarding why the new model
is superior.

4. the discussion in section 5.2 is so abstract that I don't get the insights
why the new model is better than MH. can you provide examples of spurious
structures? 

- General Discussion: The paper presents a new model for detecting overlapping
entities in text. The new model improves the previous state-of-the-art, MH, in
the experiments on a few benchmark datasets. But it is not clear why and how
the new model works better.

[Official Review · Reviewer 2 · rating 2 · confidence 3]
soundness 4 · originality 3 · clarity 3 · impact 3 · substance 4 · appropriateness 5 · meaningful comparison 2 · presentation format Poster

The paper suggests an approach based on multigraphs (several edges may link two
nodes) for detecting potentially overlapping entities.

Strengths:
The problem itself could be rather interesting especially for crossing entities
to decide which one might actually be mentioned in some text. The technique
seems to work although the empirically results do not show some "dramatic"
effect. I like that some words are spent on efficiency compared to a previous
system. The paper in general is well-written but also needs some further
polishing in some details (see minor remarks below).

Weaknesses:
The problem itself is not really well motivated. Why is it important to detect
China as an entity within the entity Bank of China, to stay with the example in
the introduction? I do see a point for crossing entities but what is the use
case for nested entities? This could be much more motivated to make the reader
interested. As for the approach itself, some important details are missing in
my opinion: What is the decision criterion to include an edge or not? In lines
229--233 several different options for the I^k_t nodes are mentioned but it is
never clarified which edges should be present!

As for the empirical evaluation, the achieved results are better than some
previous approaches but not really by a large margin. I would not really call
the slight improvements as "outperformed" as is done in the paper. What is the
effect size? Does it really matter to some user that there is some improvement
of two percentage points in F_1? What is the actual effect one can observe? How
many "important" entities are discovered, that have not been discovered by
previous methods? Furthermore, what performance would some simplistic
dictionary-based method achieve that could also be used to find overlapping
things? And in a similar direction: what would some commercial system like
Google's NLP cloud that should also be able to detect and link entities would
have achieved on the datasets. Just to put the results also into contrast of
existing "commercial" systems.

As for the result discussion, I would have liked to see some more emphasis on
actual crossing entities. How is the performance there? This in my opinion is
the more interesting subset of overlapping entities than the nested ones. How
many more crossing entities are detected than were possible before? Which ones
were missed and maybe why? Is the performance improvement due to better nested
detection only or also detecting crossing entities? Some general error
discussion comparing errors made by the suggested system and previous ones
would also strengthen that part.

General Discussion:
I like the problems related to named entity recognition and see a point for
recognizing crossing entities. However, why is one interested in nested
entities? The paper at hand does not really motivate the scenario and also
sheds no light on that point in the evaluation. Discussing errors and maybe
advantages with some example cases and an emphasis on the results on crossing
entities compared to other approaches would possibly have convinced me more.
So, I am only lukewarm about the paper with maybe a slight tendency to
rejection. It just seems yet another try without really emphasizing the in my
opinion important question of crossing entities.

Minor remarks:

- first mention of multigraph: some readers may benefit if the notion of a
multigraph would get a short description

- previously noted by ... many previous: sounds a little odd

- Solving this task: which one?

- e.g.: why in italics?

- time linear in n: when n is sentence length, does it really matter whether it
is linear or cubic?

- spurious structures: in the introduction it is not clear, what is meant

- regarded as _a_ chunk

- NP chunking: noun phrase chunking?

- Since they set: who?

- pervious -> previous

- of Lu and Roth~(2015)

- the following five types: in sentences with no large numbers, spell out the
small ones, please

- types of states: what is a state in a (hyper-)graph? later state seems to be
used analogous to node?!

- I would place commas after the enumeration items at the end of page 2 and a
period after the last one

- what are child nodes in a hypergraph?

- in Figure 2 it was not obvious at first glance why this is a hypergraph.
colors are not visible in b/w printing. why are some nodes/edges in gray. it is
also not obvious how the highlighted edges were selected and why the others are
in gray ...

- why should both entities be detected in the example of Figure 2? what is the
difference to "just" knowing the long one?

- denoting ...: sometimes in brackets, sometimes not ... why?

- please place footnotes not directly in front of a punctuation mark but
afterwards

- footnote 2: due to the missing edge: how determined that this one should be
missing?

- on whether the separator defines ...: how determined?

- in _the_ mention hypergraph

- last paragraph before 4.1: to represent the entity separator CS: how is the
CS-edge chosen algorithmically here?

- comma after Equation 1?

- to find out: sounds a little odd here

- we extract entities_._\footnote

- we make two: sounds odd; we conduct or something like that?

- nested vs. crossing remark in footnote 3: why is this good? why not favor
crossing? examples to clarify?

- the combination of states alone do_es_ not?

- the simple first order assumption: that is what?

- In _the_ previous section

- we see that our model: demonstrated? have shown?

- used in this experiments: these

- each of these distinct interpretation_s_

- published _on_ their website

- The statistics of each dataset _are_ shown

- allows us to use to make use: omit "to use"

- tried to follow as close ... : tried to use the features suggested in
previous works as close as possible?

- Following (Lu and Roth, 2015): please do not use references as nouns:
Following Lu and Roth (2015)

- using _the_ BILOU scheme

- highlighted in bold: what about the effect size?

- significantly better: in what sense? effect size?

- In GENIA dataset: On the GENIA dataset

- outperforms by about 0.4 point_s_: I would not call that "outperform"

- that _the_ GENIA dataset

- this low recall: which one?

- due to _an_ insufficient

- Table 5: all F_1 scores seems rather similar to me ... again, "outperform"
seems a bit of a stretch here ...

- is more confident: why does this increase recall?

- converge _than_ the mention hypergraph

- References: some paper titles are lowercased, others not, why?